

# Experimental Validation of a Ducted Wind Turbine Design Strategy

Benjamin Kanya[1], Kenneth D. Visser[1]

[1]Department of Mechanical and Aeronautical Engineering, Clarkson University, Potsdam, NY, USA 13699

*Correspondence to*: Ken Visser (visser@clarkson.edu)

**Abstract.** A synergistic design strategy for ducted horizontal axis wind turbines (DWTs), utilizing the numerical solution of a ducted actuator disk system as input conditions for a modified blade element momentum method, is presented. Computational results of the ducted disk have shown that the flow field for a DWT differs substantially from that of a conventional open rotor. The rotor plane velocity is higher in the ducted flow field, and more importantly, the axial velocity

component varies radially. An experimental full scale 2.5m rotor and duct were designed, using this strategy, and tested at the University of Waterloo's wind turbine test facility. Experimental results indicated a very good correlation of the data with the numerical predictions, that being a doubling of the power output at a given velocity.

## 1 Introduction

Wind energy has long been acknowledged as having the potential to supplement and even displace the carbon-based fuel

needs of our society.  The wide adoption of small wind energy, namely that with a swept rotor area of less than 200 m$^2$, has been hampered, however, by higher unit costs and lower efficiency than that of their large-scale counterparts. Studies on small turbines at Clarkson University have focused on improving their efficiency, particularly at lower wind speeds, with a focus on the key metric of cost per unit energy produced, namely $/kWh. Increasing the energy extraction for a given turbine size or reducing the manufacturing and operating costs, are both options that increase the adoption of small wind by

consumers. Other important factors that must also be considered include noise signature issues, sensitivity to wind directional changes, and issues of visibility and community acceptance.

The ducted wind turbine (DWT) concept has been fraught with controversy over the years, yet still shows promise in improving the $/kWh issue. DWTs are created by enclosing a conventional horizontal axis wind turbine with a lifting surface

geometry revolved around the rotor axis. The duct captures a larger stream tube than an open rotor, as illustrated in Figure 1. A substantial increase in velocity, exceeding even the free stream, is observed at the rotor face and the associated increase in mass flow rate increases the power output of the turbine. A properly designed DWT can improve the key areas mentioned above, leading to a much more effective small turbine design. There are, however, issues with DWTs that need to be addressed before their full potential can be realized, the foremost being the tradeoff of increased energy production against

the increased use of materials, which usually results in a higher unit cost.



This paper reports on recent experimental results that validate a synergistic design strategy of the duct and the rotor. The numerical flow field of the optimized ducted actuator disk geometry is used as the input to the blade element momentum rotor design code. In this way, the influence of the duct on the flow field of the rotor is accounted for and the rotor geometry

modified appropriately.

**2 Background**

Although studies on the potential performance gains of ducted turbines can be traced back to the 1920s, Gilbert and Foreman's (1978, 1979) extensive testing in the 1970s proposed that this occurred because the duct reduces the pressure

behind the turbine, relative to that behind a conventional wind turbine, causing more air to be drawn through. They suggested that they could have a performance efficiency of $Cp = 1.57$, defined as

$$C_p = \frac{\text{Power Extracted}}{\text{Power in Wind}} = \frac{2P_{turbine}}{\rho AV^3}$$

where A = rotor area. The maximum $Cp$ for an un-ducted open rotor is 0.593, commonly known as the Betz limit.  This leads to the definition of an 'augmentation ratio' of r = 2.65 where

$$r = \frac{C_{p,Ducted\ Turbine}}{C_{p,Betz}}$$

Hanson (2008) suggested that it is the lift generated by the shroud, as shown by de Vries (1979), that induces an increased mass flow through the rotor, resulting in an increase in the power coefficient proportional to the mass flow. Although one might surmise that, via this increased mass flow rate and velocity, a DWT can exceed the Betz limit, this is incorrect, because a much larger stream tube has been captured, and the assumptions applied to the open rotor case do not apply. Unfortunately, over the years, claims by inventers that they have 'beaten Betz' have only served to give DWTs a bad

reputation.

Many studies have investigated the feasibility and associated augmentation factors seen in DWTs in an effort to further their development (Hu et al, 2008; Igra, 1976, 1984; Hansen et al, 2000; Werle and Presz, 2008; van Bussel, 2007; Oman et al, 1977; Leoffler, et al, 1978; Ohya et al, 2002, 2008; Politis and Koras, 1995; and Jamieson, 2009) with the largest prediction

of 7 by Badawy and Aly (2000), however conclusions have been quite varied. Werle and Presz (2008) used fundamental



momentum principles and concluded that the possible augmentation factor could only approach 2, and that earlier studies had incorrect assumptions, leading to overly optimistic predictions. Hansen's (2000) viscous CFD results predicted ideal $Cp$ values approaching 0.94, and an augmentation factor of 1.6. He also indicated that if the duct geometry could be made to keep the flow attached, the augmentation factor could be improved further.

A review article by van Bussel (2007) substantiates the above arguments regarding mass flow and indicates that the increase of the mass flow, and thus the augmentation ratio, is proportional to the ratio of the diffuser area to the rotor area. van Bussel concludes that the amount of energy extracted *per unit volume of air* with a DWT *remains the same* as for a bare rotor, but since the volume of air has increased, so has the total energy extracted. He noted that Cp values above 1,

corresponding to augmentation ratios on the order of 2, are achievable with diffuser to inlet area ratios on the order of 2.5. In addition to the experimental data, he reported on the effect of reducing the back pressure, which can also have a profitable effect on the performance.

This potential increase in power generation has continued to drive DWT research, however no commercial design has been

able to realize these augmentation factors and no commercially viable DWT has been successful. A good example of this type of failure is seen in the Vortec 7 from New Zeland in Figure 2a. (Phillips et al, 2003; Windpower Monthly, 2001). A more recent example is that of the demise of Ogin (Boston Globe, 2017) in Figure 2b.

Perhaps the most promising experimental field results have been that of Ohya at Kyushu University in Japan on ducted

turbines with a brim at the trailing edge. (Ohya, 2014). Experimental data has been obtained on several units, including 500W, 3 kW, 5kW and 100kW units, with measured power coefficients approaching a $Cp = 1.0$. He has also reported no appreciable increase in the noise levels generated by the turbine while running.

Recent results of the synergistic design of the duct and rotor flow field at Clarkson University have indicated two key design

aspects. First, the presence of the duct modifies the axial velocity at the rotor, as shown in Figure 3a (Jedamski and Visser, 2013) from a nominally uniform distribution to one with a radial variation. Second, moving the rotor to a location aft of the throat, Figure 3b, provides an increased power output for a given duct geometry (Visser, 2016, Bagheri-Sadeghi et al, 2017). Most rotor designs seek to exploit the high velocity at the throat of the duct, however the presence of the rotor modifies the velocity where it is stationed, and more power can be extracted from the design by moving the rotor aft. The optimum blade

design for the rotor is not that which would be required of an open rotor, but is different in planform shape and twist, due to the presence of the flow field generated by the duct. Venters et al (2017) has also indicated $Cp$ values, based on the duct area, of greater than 0.593, possibly pointing the way for a wind energy extraction device that is more efficient than a turbine of equal diameter.



.

Perhaps the most enticing aspect of the DWT concept is the potential for increased energy production in lower speed wind regimes, opening up many more areas to a viable distributed wind energy solution. Based on the above promising results, this investigation was undertaken to experimentally validate the synergistic design strategy.

**3 Investigative Methods**

In order to make a comparison of the experimental data to existing available turbines, a rotor diameter of 2.5 meters was selected for the design to compare to a commercially available turbine, the Excel 1 by Bergey Windpower (2017). The Excel 1, illustrate in Figure 4 is a 2.5 m diameter open rotor with a maximum output of 1.2 kW. The blades are constant chord and untwisted.   The current 2.5 m prototype was optimized specifically for the ducted turbine environment.   The test plan

focused first on evaluating the open rotor design against the Bergey open rotor, and then examining the effect of the duct. Details of the numerical design are presented below followed by the experimental methods overview.

**3.1 Numerical Approach**

The numerical design strategy used a two part scheme. First, the flow field of the duct with an actuator disk was determined using the Navier-Stokes solver FLUENT. Details of the methods employed can be found in Bagheri-Sadeghi et al (2017) and

an example is shown in Figure 5a. From this solution, the axial velocity field was then extracted, Figure 5b, and used as an input for Clarkson's in-house Blade Element Momentum (BEM) code, *mRotor*. The rotor design in *mRotor* uses a fairly standard BEM strategy by Glauert to determine the optimum rotor shape (Kanya and Visser, 2010). Figure 6 illustrates the typical forces and velocities, including the axial interference factor 'a'. A second piece of information from the numerical code, the thrust coefficient at the rotor, $C_{T,rotor}$, was also extracted and the axial interference factor, a, was determined for

input to *mRotor* as:

$$a = \frac{C_{T,rotor}}{4 + C_{T,rotor}}$$

where

$$C_{T,rotor} = \frac{2\Delta P}{\rho V_{rotor}{}^2}$$

It is important to note that when designing a rotor, one is often, as is the case here, designing for a given generator, as there is only a discrete set available. In this case, as the generator was a 1.8 kW unit, requiring about 45 Nm at the 480 RPM rated speed, it was critical that the rotor could deliver this torque at that RPM. This usually means a compromise with the ideal aerodynamic tip speed ratio, TSR, that being the ratio of the tip speed of the blade to that of the oncoming velocity. In addition, it was desired to use a simple airfoil, either a flat plate or a curved flat plate like the GOE417a, as shown in Figure



7, because of the lower RPM. Airfoils, like these, work well in low Reynolds number environments and an additional goal was a cheaper manufacturing strategy. Since the optimum TSR is a function of the airfoil performance (Kanya and Visser, 2010), the selection of a less than optimal airfoil can be a better choice for a lower TSR, rather than having a better performing airfoil operating in an off-design condition.

The design TSR was set to 4 and the blade number selected was 3, despite the slightly higher aerodynamic gain potential with added blades. The presence of the duct mitigates the tip losses to some extent and when coupled with a constraint on the budget, pushed the design to a 3 bladed configuration. Figure 8 illustrates the final design with a solidity of 9.8%

**3.2 Experimental Setup**

The Clarkson DWT was designed as a prototype for the NEXUS-NY competition funded by the New York State Energy and Research Development Authority. Since the goal was to undergo wind tunnel testing, many of the requirements of a commercially viable turbine, such as a yaw bearing and weatherproofed materials, were not required, and were not included in this prototype build.

The duct was constructed out of EPS foam covered in a StyroSpray polymer as shown in Figure 9. The exit diameter of the duct was 3.3m leading to a ratio of the exit area to the rotor area of 1.74. The ratio of the duct length to rotor diameter was 0.25. The rotor blades, one of which is shown in Figure 10, were numerically milled from aluminium to match the BEM design with a mounting boss that enabled them to be bolted to the hub. The rotor was attached to a 1.8 kW radial flux

permanent magnet generator from Ginlong, model GL-PMG-1800. The 3 phase AC generator output was rectified through a Microsemi MSD52 Glass Passivated Three Phase Rectifier Bridge and the DC measured with a BK Precision 8522 Programmable DC Electronic Loads Unit.

Wind tunnel testing was conducted in November 2016 at the University of Waterloo Wind Turbine Test Facility in

Waterloo, Ontario, Canada. An exterior view of the facility highlighting the 6 external fan/blower array is shown in Figure 11. The turbine was first tested in an open rotor configuration as illustrated in Figure 12. After a suitable amount of data was acquired, including repeat runs, the duct was attached to the turbine stand, Figure 13, and the runs repeated. The duct was then removed and the open rotor tests repeated. A ¾ view of the test setup including the upstream sonic anemometer is shown in Figure 14.


Tunnel velocity was varied in 1m/s increments to a speed that caused to generator to slightly exceed the maximum rating of 1800W. Velocity was measured upstream of the rotor with a sonic anemometer. At each velocity test point, the resistive load on the turbine was varied until the power output, namely the product of the voltage and current, was the greatest. RPM was





recorded using a magnetic bicycle switch. Since there was no electronic braking circuitry or dump load, for each test point the resistive load was first increased (Ohms were lowered), followed by an increase in tunnel speed, to prevent a runaway condition.

## 4 Results

5 The results of the wind tunnel testing are discussed below, with a focus on the power performance. Power results are followed by additional observations on the energy production implications and the behaviour of the flow field. During the test, a hand held anemometer was positioned at various locations in the flow field to gain some understanding of the velocity field near the rotor, as the duct did not have pressure taps. It was observed that the velocity at the duct inlet face was about 10-15% above the upstream reference value across the entire speed range.

10 **4.1 Power Performance**

Figure 15 is a plot of the wind tunnel results. The black triangles represent the published power curve of the Bergey Excel 1 turbine. The solid filled circles are the data for the Clarkson open rotor configuration. These open rotor tests were conducted with the hub pitch set to 45° (nominal) and then 46° as an after check. It can be seen that the performance of the current rotor is slightly better than the Bergey, and is to be expected. The Bergey uses an untwisted constant chord blade, while the 15 Clarkson blade has an optimum twist and planform distribution. Both sets of data sit below the upper theoretical limit for a 2.5 meter open rotor, the Betz limit, as they should, which is denoted by the solid line.

The ducted data is marked by the open circles in Figure 15. It was observed that the power output substantially increased with the presence of the duct. At 9 m/s, for instance, the Bergey generated about 700W, while the Clarkson open rotor 20 configuration puts out about 925W. With the duct installed, the turbine output was increased to about 1880W. Thus, the duct increased the power output to approximately twice the un-ducted configuration. Unfortunately, 9 m/s was the highest speed that could be tested, as the generator was producing above the rated output at that speed and could not be increased further.

Perhaps most interesting, however, are the two light grey dotted lines bracketing the data in Figure 15. These are the 25 predicted output curves from *mRotor*, with and without the tip loss corrections included, indicating that the ability to synergistically design and predict the performance of the ducted turbine, with the numerical simulation input to the *mRotor* code, was validated by the wind tunnel data. The last curve on the plot, denoted by the heavy dashed line, is the numerical prediction by the actuator disk model in FLUENT, namely the solution used to generate the input velocity field profile for the *mRotor* design. The power predicted represents the possible upper limit to the turbine performance. For more details on 30 the numerical input solution, see Bagheri-Sadeghi et al (2017).

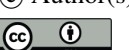



The *Cp* values, based on the rotor area, were also calculated for the data and are presented in Figure 16. The Bergey reached a peak value of about *Cp* = 0.37 at 7 m/s. The Clarkson open rotor configuration remains fairly constant at about at *Cp* = 0.41 from 5 – 9 m/s. The ducted configuration generated values of about *Cp* = 0.85 – 0.90 over the same range. It is important to note that although these values lie above the Betz limit line, this does not in any way indicate the theoretical

limit has been exceeded. That limit applies only to an open rotor. A ducted turbine captures a larger stream tube and one would need to determine what amount of power was in that stream tube to get a sense of the 'efficiency' of the ducted turbine. One possibility is to non-dimensionalize the data by the exit area of the duct, but that is best left to a separate discussion. Nonetheless, the power output was significantly increased and, depending on the increased cost of the duct, represents a viable way to increase the power output of a given rotor size.

A closer look at the data revealed that the torque was consistently lower than what was predicted, and conversely the RPM values were higher, at each of the operating points. At 1800W, for instance, there should be around 45-50Nm of torque at 350 rpm, however an 450-500 rpm was observed implying the torque to be only 35-37 Nm. A check on the tip speed ratio values indicated they were also high, above 6, when they were designed to be below 5. Overall, the torque was consistently

30-40% low across the speed range, while the RPM values were about 35% higher than expected.

A series of additional tests were run at a wind speed of approximately 3.3 m/s to vary the pitch angle of the blades and observe if the power output could be improved. Table 1 lists the geometries tested and the output.

**Table 1: Impact of Hub Pitch Angle on Ducted Configuration**

| Hub Pitch Angle | Power Output at 3.3 m/s |
|---|---|
| Decreased (42°) | 83 W |
| Nominal (45°) | 93 W |
| Increased (48°) | 92 W |


At each condition, the load was varied until the maximum power was obtained. It appeared that the nominal hub pitch angle of 45°, as predicted by *mRotor*, was close to, if not the correct setting predicted by *mRotor*. It also seems that a variation of a degree or two would not impact the performance significantly.

**4.2 Energy**

Although the efficiency of the power curve is an important characteristic of a turbine's performance, it is the energy generated that is the most important. Society pays for energy, and power is simply the rate at which it is used. Table 2 contains the Annual Energy Produced (AEP) for each configuration based on the typical 5 m/s Rayleigh probability distribution (PDF), Figure 17, and their respective power curves. As can be seen, if the ducted 2.5 m rotor was limited in



power output to that of the Bergey Excel 1, namely 1.2 kW, the Clarkson turbine would still produce twice as much energy. Fitting the rotor to a 3.5 kW generator would increase the energy output to approximately 2.7 times that of the Excel 1. This highlights the inadequacy of simply rating a turbine by the power it produces at a given speed, typically 11 m/s for small turbines. It is the AEP at a given wind PDF that should be the defining quality.

**Table 2: AEO Comparison of 2.5 m Rotors Using Standard Rayleigh Distributions (kWh)**

| Turbine | Vavg = 4 m/s | Vavg = 5 m/s |
|---|---|---|
| Bergey Excel 1 | 1055 | 1923 |
| Clarkson (1.2 kW) | 2497 (2.4x Excel) | 3931 (2.0x Excel) |
| Clarkson (3.5 kW) | 2845 (2.7x Excel) | 5263 (2.7x Excel) |

The observed improvement results primarily from an increased performance output of the ducted turbine at low speeds. This can be inferred from the power plot of Figure 15, but what isn't so obvious is the role the PDF of the average wind speed plays. Figure 17 illustrates that the majority of time, for the 5 m/s PDF, the wind is blowing below 6 or 7 m/s. This is why

15  the improved low speed performance from the duct is so effective on the energy extracted. If a wind regime with the average speed of 4 m/s is now considered, the results are even more illustrative, as noted in Table 2. At 4 m/s the 1.2kW ducted configuration now generates 2.4 times more energy than the Bergey. Note, however, that the 3.5 kW advantage is about the same, indicating perhaps that the generator needs to be sized according to the local wind regime as well.

20  **4.2 Flow Field Issues**

In an effort to understand the flow field a little better, a lower quadrant of the duct was tufted to examine the surface flow as shown in Figure 18a. The tufts indicated substantial separation behind the support struts and near some regions of the trailing edge. There were also regions of fluctuating separation from the mid-chord of the duct aft to the trailing edge. Figure 18b illustrates an example of this region. The observed fluctuations in the surface flow, causing periodic separation

25  and reattachment, may have been in part due to the upstream blade passage, but was clear that there were also other frequencies involved. Although only subjective observations were made, a time averaged separation of more than 50% could be seen in some regions. In light of this, it is surprising the overall performance was as good as it was and suggests that improvements can be made.



### 4.3 Final Considerations

An argument could be made, and has been made to the authors, that rather than using a duct, one could simply increase the size of the rotor and accomplish the same increase in energy output, and this is absolutely true. In fact, the material required for the rotor would likely be less than that needed for the duct. For the case at hand, assuming the power is proportional to

the area, all else being equal, one would need an increase of a factor of 2.7 times the area, from d=2.5m (8.2ft) to d=4.1m (13.5 ft), or an increase in diameter of over 50%, not an insignificant increase in the blade size, but certainly doable. Note that the duct exit diameter for the prototype was 3.3 m, 20% smaller than the required rotor size increase would have to be.

It is important, therefore, to highlight the additional advantages of a ducted turbine. For a given energy requirement, the

DWT can be made smaller. Conversely, for the same size rotor, the DWT would produce more energy. Alternatively, one can utilize a lower tower to achieve the same AEP. Most importantly, however, is that a lower wind speed regime can be utilized to provide the same energy when using a ducted tower and this is, arguably, the most significant factor, for it opens up many more areas to wind evergy.

From a design point of view, one can use a smaller generator to achieve the same output AEP and this in turn reduces the cost. A smaller generator also reduces the mast head weight and ease of installation, possibly alleviating the need for a crane. More subjective arguments, such as redundancy, can be debated, but if two turbines can produce the same amount of energy as one turbine, and the cost is the same, the advantage is quite obvious. The current design uses a low RPM generator, which helps alleviate noise issues and the duct helps a bit in this area as well. Issues of ice throw from the blades can be mitigated

with the duct, as well as blade throw, should a failure occur. Although a contentious topic, the impact on avian life can be argued to be lessened as birds can always see the duct and have even been seen to perch on the top of ducts in the past.

A final consideration is the key metric mentioned previously: cost per unit energy. Despite all the well-mannered intentions of those seeking a greener method of energy generation, it is this $/kWh, particularly over the life of the unit, also known as

the Levelized Cost of Energy (LCOE), that will be the defining factor for the Ducted Wind Turbine. Whether or not a viable solution to this issue exists is a question that is still to be answered.

### 5 Conclusions

The primary conclusions of this study were:


1. The ability to predict the performance of a ducted turbine using a synergistic combination of a numerical flow field model as the input to a blade element momentum model was validated.

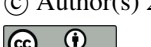


1. A prototype 2.5 m ducted rotor was tested and exhibited the potential for generating twice as much energy as a conventional open rotor, indicating experimental Cp values of 0.8 to 0.95, almost three times that of small commercially available turbines.

3. The use of a ducted turbine configuration can reduce the size of the required generator and possibly the weight of the entire turbine at the mast head.

## 6 Author Contributions

10   Ken Visser and colleague Paul Pavone designed the experimental apparatus and Ken Visser carried them out. Ben Kanya developed the model code and performed the simulations. Ken Visser prepared the manuscript.

## 7 Competing Interests

The authors declare that they have no conflict of interest.

## 8 Disclaimer

15   This work is original and has not been published elsewhere.

## 9 Acknowledgements

The authors would like to thank the many individuals who have contributed to the project including: Brian Helenbrook, Nojan Bagheri-Sadeghi, Paul Pavone, Devon Jedamski, Steve McCauliff, Hebron Yam, Stuart Wilson, Cameron Gibb,
20   Alison Davis, Mike Valleau, and of course Jacob Weller from the Clarkson University machine shop facilities. We would also like to thank Professor David Johnson from the University of Waterloo for the use of his facility and the help of his students: Leif Falk, Michael McKinnon, Farid Samara.  Finally, we are very grateful for the funding support of this project from the New York State Energy and Research Development Authority (NYSERA) through NEXUS-NY.



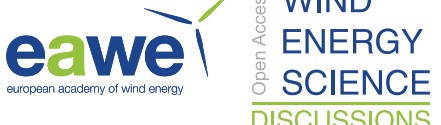

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



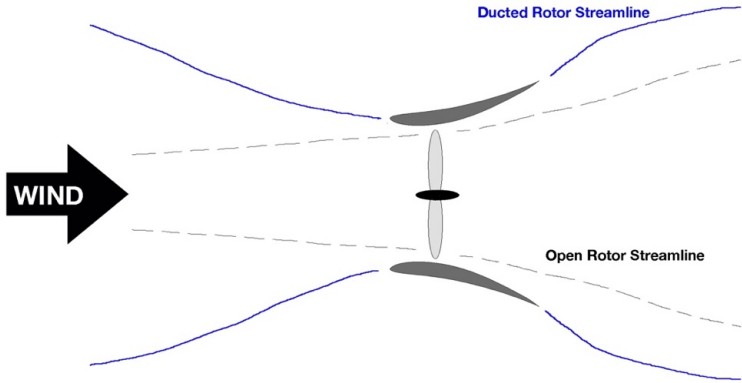

**Figure 1: Stream Tube Capture Regions for Open Rotor and a Ducted Rotor Turbines.**

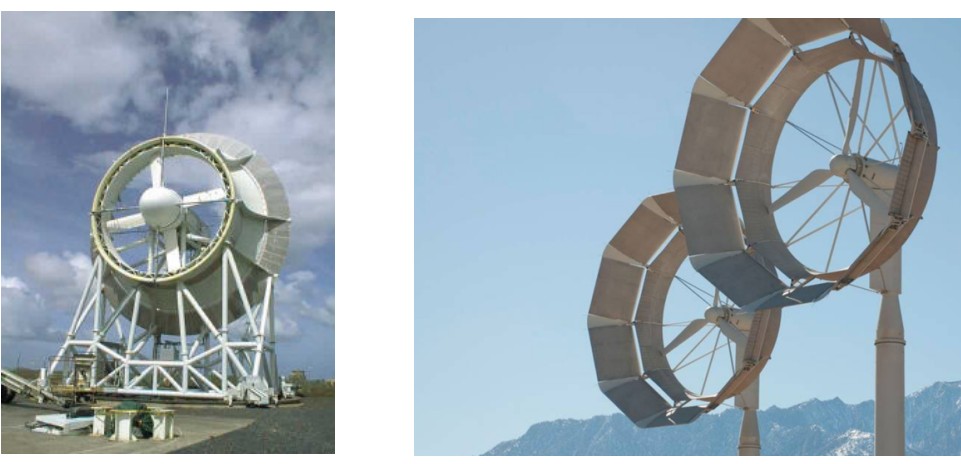

5    **Figure 2: Commercial Attempts at Large Ducted Turbines a) Vortec 7  b) Ogin (FloDesign)**



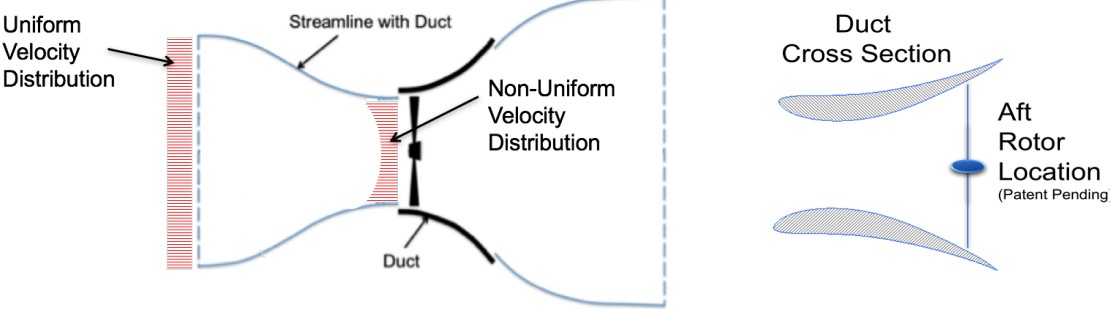

Figure 3: Key Design Aspects for the Clarkson Ducted Turbine a) Non-uniform Velocity Distribution  b) Aft Rotor Location.

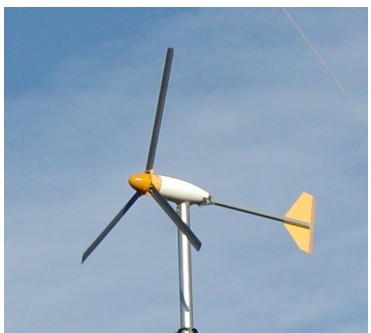

5    Figure 4: Bergey Excel 1.

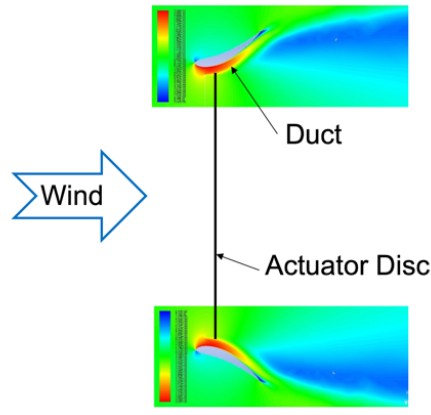

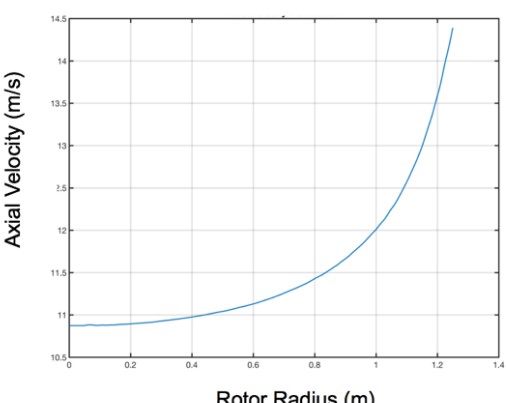



**Figure 5:  Numerical Duct Results  a) Flow Field Solution   b) Extracted Velocity Profile.**

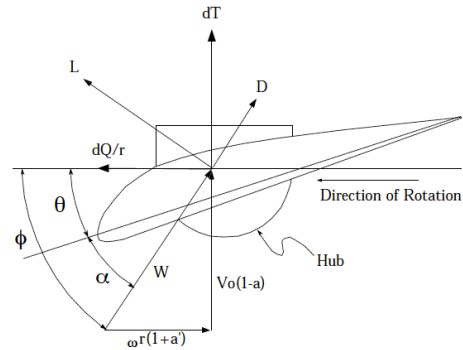

**Figure 6: Blade Element Momentum Forces and Velocities.**

**Figure 7: Rotor Blade Airfoil.**

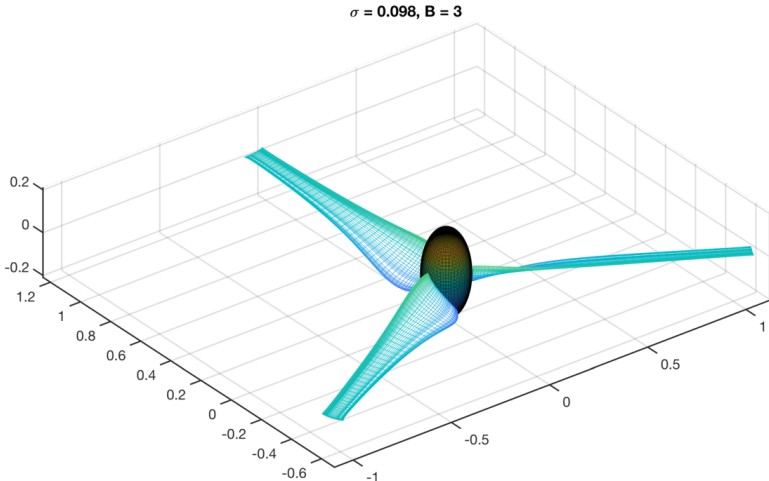

10      **Figure 8: Blade Geometry Designed with *mRotor*.**




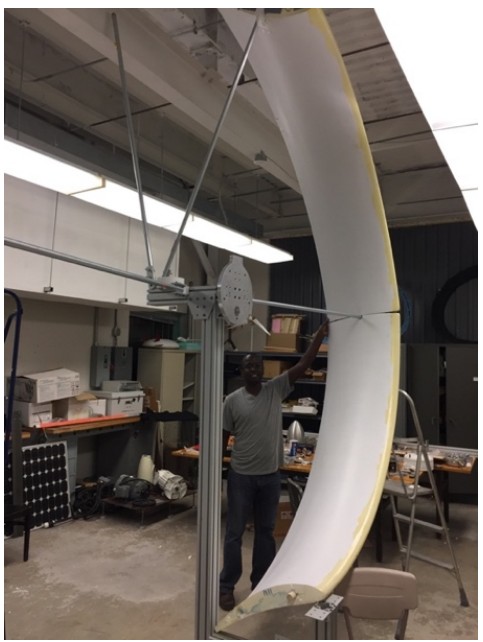
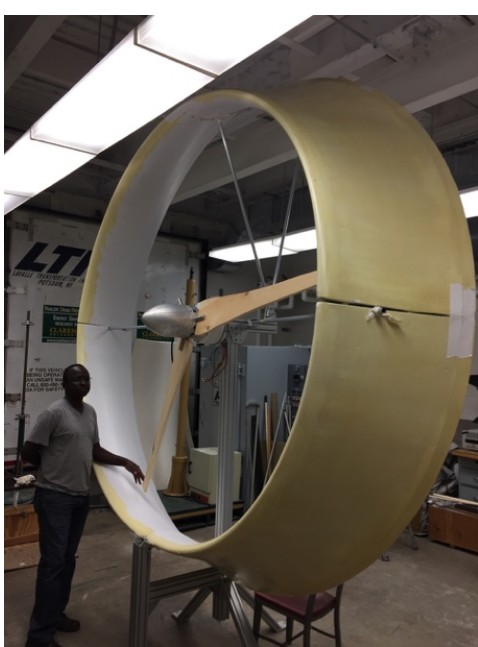

**Figure 9: Construction of Duct at Clarkson University.**

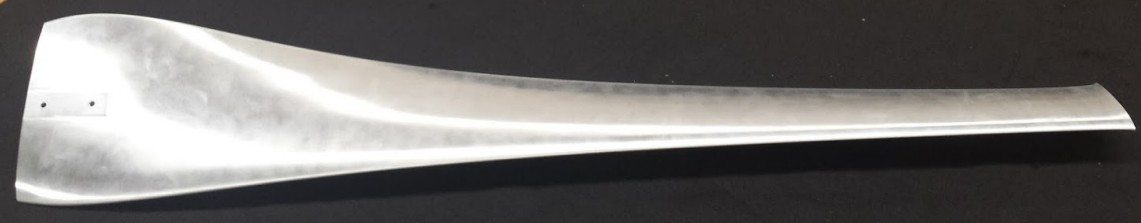

**Figure 10: Aluminium Rotor Blade.**




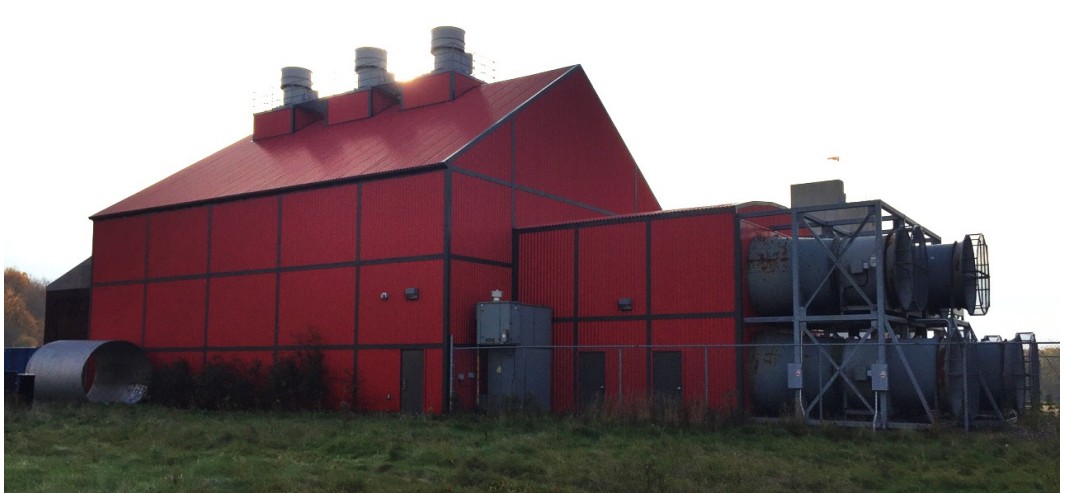

**Figure 11: University of Waterloo Wind Turbine Test Facility.**

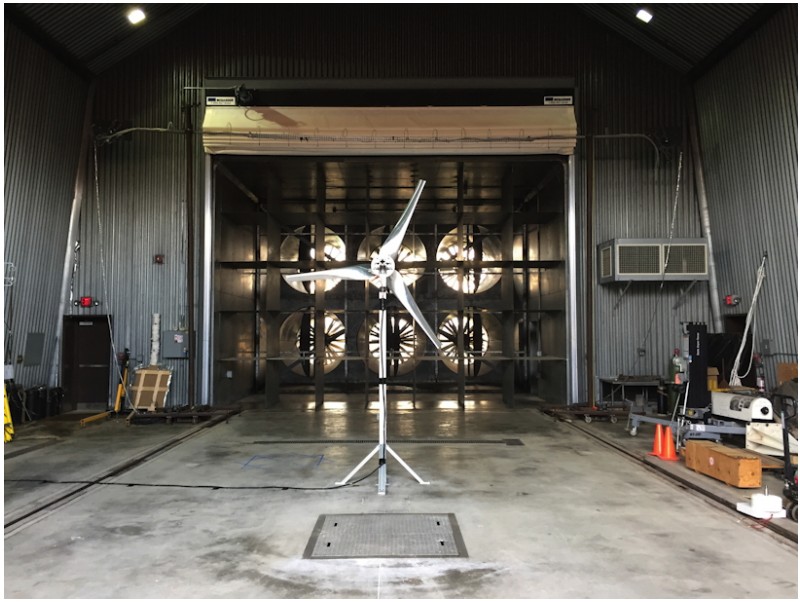

5   **Figure 12: Open Rotor Turbine Setup in Waterloo Wind Tunnel.**




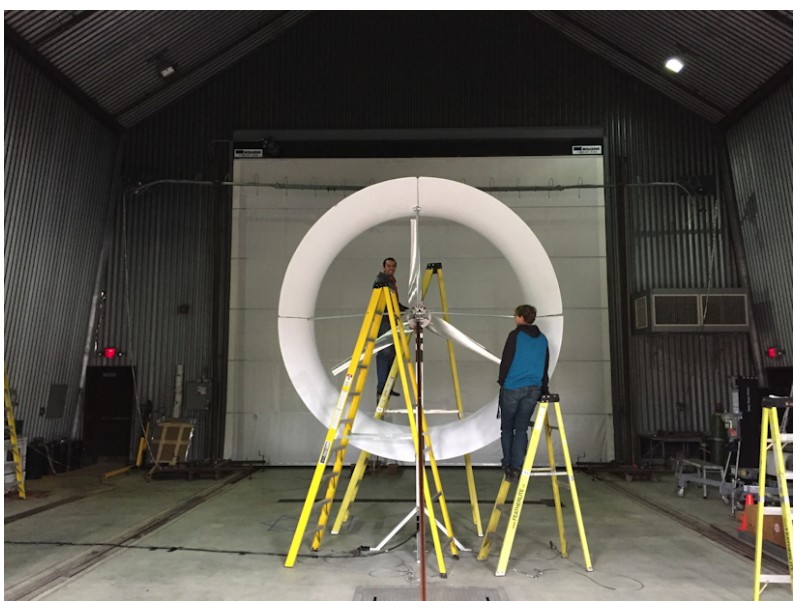

**Figure 13: Ducted Rotor Preparation for Testing.**

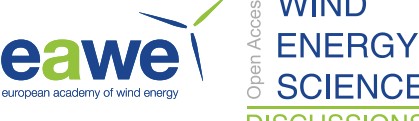

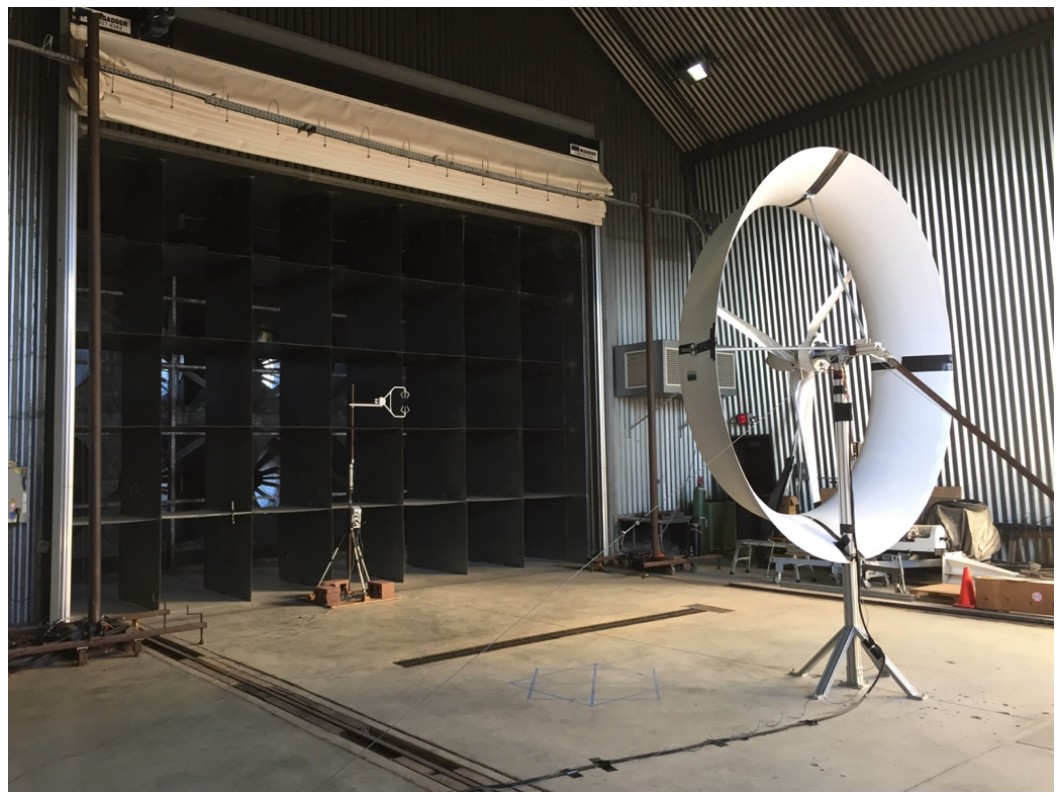

**Figure 14: Ducted Rotor Under Test Conditions.**

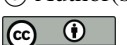



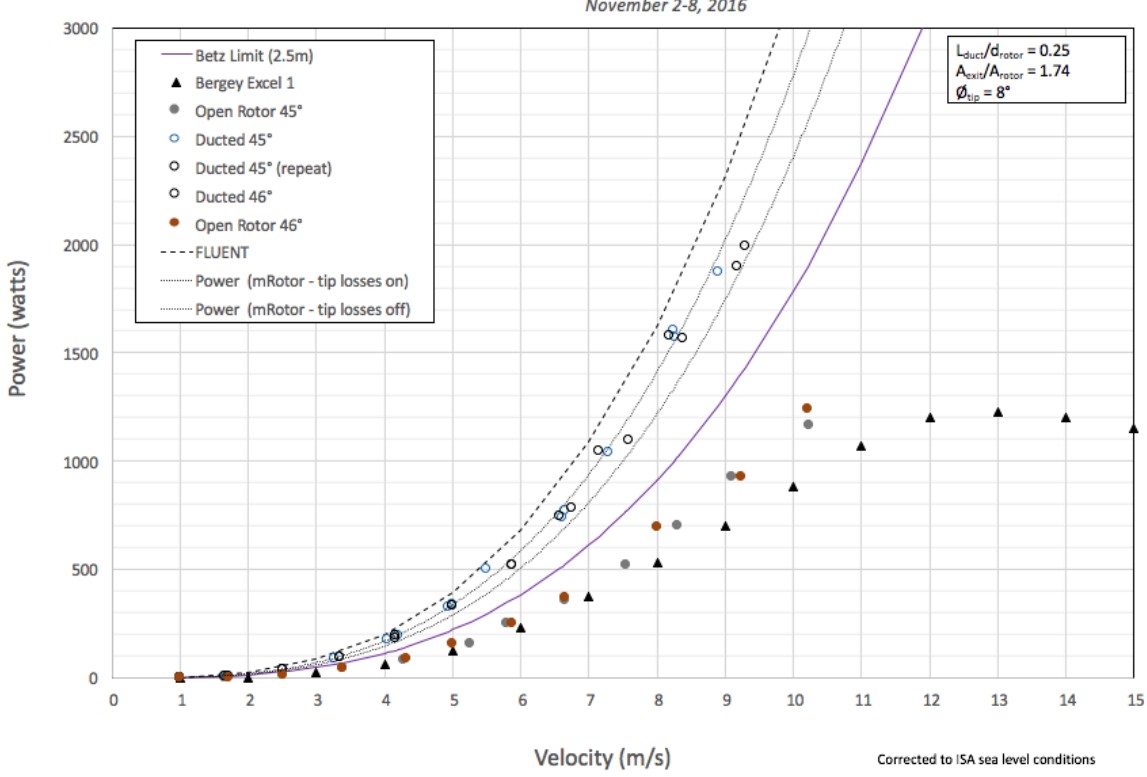

**Figure 15: Wind Tunnel Power Curve Performance.**

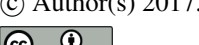



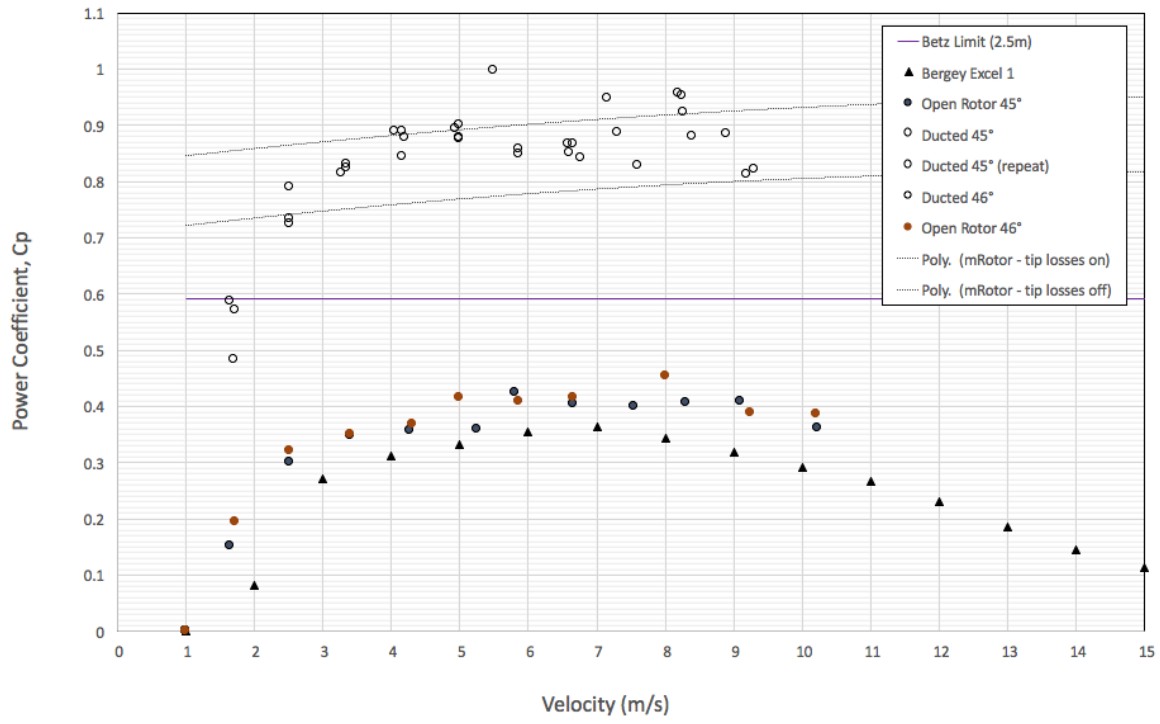

Figure 16: Experimental Cp Values (based on rotor area).





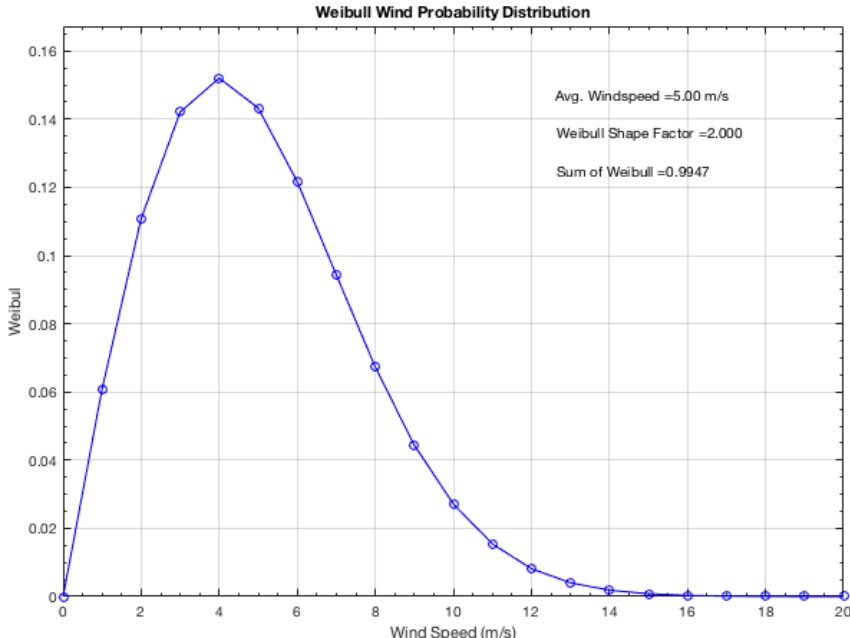

Figure 17: Rayleigh PDF for a 5 m/s Average Wind Speed .

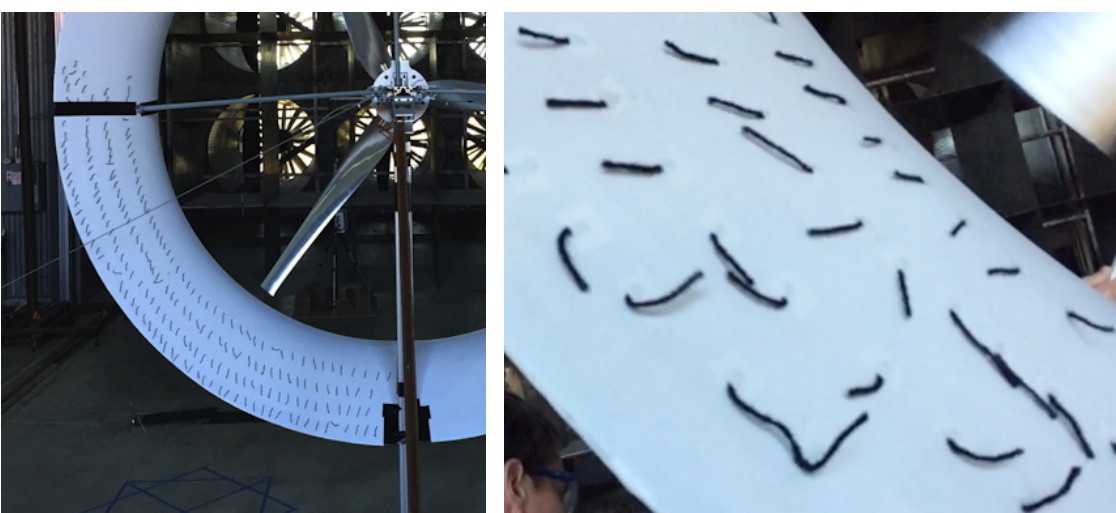

5    Figure 18: Tufting on the Ducted Surface.