# Peer review of "Experimental Validation of a Ducted Wind Turbine Design Strategy"

_Wind Energy Science, 2017_

## Referee Comment (RC1) · Anonymous Referee #1 · 24 Dec 2017

WES2017

Title : Experimental Validation of a Ducted Wind Turbine Design Strategy Author(s) : B. Kanya and K.D. Visser

Referee's Comments :

This paper describes an experimental and numerical study of the flow around a ducted wind turbine. For the wind tunnel experiment, they used a prototype wind turbine made by Clarkson University with the rotor diameter of 2.5m and duct diameter of 3.3m, it means the flow around the blades is enough in high Reynolds numbers. They used a DNS provided by Fluent not using any turbulence model to analyze the flow around the turbine adopting an actuator disk model. The results obtained are reasonable and the

power output coefficient Cp values are comparable with other values reported in the past. The results obtained confirm the findings which were already reported from other research using wind tunnel experiments and CFD. However, the referee wonder what are the new findings and what are the inventions in the present research.

For the overall evaluation, the present paper gives relatively good experimental and numerical results and suitable discussion. However, a couple of discussions seem to be insufficient. The referee described those problems below in the detailed points. It seems to the referee that a further improvement and revise still remain for the present paper. If the authors can give further information, the present paper should be accepted for the WES under this content.

Major points:

For the numerical method, what kind of grid type did the author employ? What is the resolution around the turbine? Did the authors pay attentions to the Reynolds number effect and the grid resolution dependence to make clear the flow characteristics around the ducted wind turbine which shows flow separation and reattachment inside the duct, and vortex shedding from the duct. The authors discussed the surface flow feature near the exit of the duct in the 4.2 Flow Field Issues. It strongly suggest that the flow around the ducted turbine both inside and outside of the duct are highly unsteady, unstable and turbulent flows. The reviewer cannot understand the accuracy of CFD presented in this paper.

For the wind tunnel experiments, what is the uniformity of approaching wind in the wind tunnel of the University of Waterloo? What is the turbulence intensity and its uniformity? What is the blockage ratio? These are the fundamentals of the specification of a wind tunnel. It seems to the referee that the experimental results from the combination of the present wind tunnel and a wind turbine model, there are a little blockage effect in the results.

For the ducted wind turbine prototype employed here, the authors should describe the

figure of the curved shape of the duct with the inlet diameter, throat diameter and exit diameter

The referee recommends that the author should evaluate the modified Cp which adopts the maximum duct area, i.e., the projected area of the duct as the reference area. For reference, please check a paper of Energies 2010, 3, 634–649; doi:10.3390/en3040634 "A Shrouded Wind Turbine Generating High Output Power with Wind-lens Technology"

Furthermore, it is necessary to correct the following points.

Detailed points

1. p. 4, l 18-20; the author should explain "a" in more detail. And in also Figure 6, what are $\omega r(1+a')$, Vo(1-a). Please describe the definitions of all the symbols in the present paper.

2. Figure 5 b); the scales are written in so small letters

Please also note the supplement to this comment:
https://www.wind-energ-sci-discuss.net/wes-2017-54/wes-2017-54-RC1-supplement.pdf

---

## Referee Comment (RC2) · Anonymous Referee #2 · 26 Dec 2017

Adding a duct (or diffuser) to a wind turbine to increase the mass flow and power is a simple modification but it provides significant theoretical and computational challenges. There is no shortage of theories for diffuser augmentation, but little detailed experimental knowledge to guide their development. This paper is a very welcome addition to our experimental knowledge. One of the problems with testing diffuser augmented wind turbines (DAWTs) is that the diffuser increases the turbine cross sectional area and so wind tunnel blockage issues can become important. The tests reported in the paper were done in the large University of Waterloo facility which is, presumably, why "blockage" is not mentioned. The most important results are those shown in Figures 15 and 16 where a significant augmentation of power over the bare turbine is shown. I have several reservations about the experiment. The first is that no detailed description of

the rotor or the diffuser is given so that future developers of DAWT models do not have the necessary information to test their theories. Secondly, the tests used a permanent magnet generator for which the manufacturer provides the efficiency curve for only one load. The experimental procedure of varying the load in search of the maximum power point is a sensible one, but it does not address the issue of the varying turbine efficiency. This is required to determine the extracted aerodynamic power, which is the usual target of the theories mentioned above.

---

## Author Comment (AC1) · 7 Feb 2018

Dear Referee #1

Many thanks for your comments and advice.

I have addressed all your questions below on behalf of my co-author as well, however I have not attached a modified manuscript yet. Please let me know if this is sufficient and the answer to a couple questions below.

1. What are the new findings and what are the inventions? The primary finding is that a computational design strategy, involving the evaluation of a numerical solution of the ducted flow field as an input to the design code for the rotor itself, has been experimentally validated. This enables one to design a ducted turbine to a different

size with a much higher degree of confidence in the real world performance. It also enables us to achieve higher Cp values.

2. For the numerical method, what kind of grid type did the author employ? The grid had a boundary layer mesh with a growth rate of 1.1 and the first mesh point was set at yˆ+≈1. The boundary layer thickness was calculated as a function of Re_c for each case and enough inflation layers were used to span the entire boundary layer. A layer of quadrilateral elements covered the actuator disc which was used to model the turbine in our 2D axisymmetric model. There was a refined unstructured triangular grid around duct which was surrounded by a large structured quadrilateral grid covering further the upstream and downstream of the actuator disc. The rest of domain was meshed with unstructured quadrilateral elements (please see the companion article under review for WESC by Bagheri-Sadeghi et al., "Ducted Wind Turbine Optimization and Sensitivity to Rotor Position" for further details)

3. What is the resolution around the turbine? The actuator disc is covered with 200 elements (i.e. for the 2.5m rotor each element in our axisymmetric model covered 6.25 mm)

4. Did the authors pay attentions to the Reynolds number effect and the grid resolution dependence to make clear the flow characteristics around the ducted wind turbine which shows flow separation and reattachment inside the duct, and vortex shedding from the duct? As mentioned above the boundary layer thickness and y+ was calculated for each case to make sure the most accurate results that one can obtain from a RANS CFD solution can be obtained. The k-$\omega$ SST turbulence model was utilized, which among the two-equation turbulence model gives better prediction of flow separation.

5. The authors discussed the surface flow feature near the exit of the duct in the 4.2 Flow Field Issues. It strongly suggest that the flow around the ducted turbine both inside and outside of the duct are highly unsteady, unstable and turbulent flows. The

reviewer cannot understand the accuracy of CFD presented in this paper. Although the predictions of power output from the CFD model agreed very well with the experimental results, as you suggested there was a difference between CFD predictions and experimental results. The CFD model showed no flow separation whereas flow separation was observed in the experimental tests. This could be due to simplifications of the CFD model like using a 2D axisymmetric model where the turbine was replaced with an actuator disc, or the limited accuracy of two-equation turbulence models. It could also have resulted from the differences of the manufactured model from the CFD model; from the final geometry of the duct, to the actual physical supporting structures used for the duct, and the influence of the individual rotor blades (as opposed to a uniform disc) which affect the flow field. Lastly, the proximity of the ducted wind turbine to the floor could have aggravated the flow separation.

6. For the wind tunnel experiments, what is the uniformity of approaching wind in the wind tunnel of the University of Waterloo? The uniformity of the wind was not explicitly mapped. The flowfield is generated by a bank of 6 100 hp fans with independent variable speed control. Data was acquired with a single location sonic anemometer. At a given velocity setting the flowfield was sampled with a hand held anemometer and compared to the single point. The average variation of the incoming flow to the turbine varied by a +/- 1-2%.

7. What is the turbulence intensity and its uniformity? The turbulence of the flow filed was sampled over the range of the incoming velocities and was found to vary over the range of 5-10%

8. What is the blockage ratio? The blockage ratio, based on the projected frontal area of the duct and the rotor, for a stationary rotor, was 4.1%. If the rotating rotor is considered to act as a solid blockage, the ratio would be 11.7% No blockage corrections were made.

9. For the ducted wind turbine prototype employed here, the authors should describe

the figure of the curved shape of the duct with the inlet diameter, throat diameter and exit diameter We can include a figure of the detailed geometry or a text description, namely that the airfoil was an Eppler 423 with a chord of 0.6 m with inlet, throat and exit diameters of x, x, and x. Which would the referee feel to be more appropriate?

10. The referee recommends that the author should evaluate the modified Cp which adopts the maximum duct area, i.e., the projected area of the duct as the reference area. For reference, please check a paper of Energies 2010, 3, 634–649; doi:10.3390/en3040634 "A Shrouded Wind Turbine Generating High Output Power with Wind-lens Technology" Yes, the Cp values are scaled by the rotor area. Scaling by the projected are of the duct is, perhaps, a more 'fair' evaluation of the data compared to a conventional turbine. In the case here, the exit diameter is 3.3 m and the rotor diameter is 3 m. Hence the Cp values would all be scaled by the ratio 3ˆ2/3.3ˆ2 or a factor of 0.826. Would the referee prefer the data to be shown in a separate plot or on the same plot in Figure 16?

Detailed points 1. p. 4, l 18-20; the author should explain "a" in more detail. And in also Figure 6, what are $\omega r(1+a')$, Vo(1-a). Please describe the definitions of all the symbols in the present paper. A more detailed explanation of a and a' will be put in the manuscript.

2. Figure 5 b); the scales are written in so small letters The plot scales will be increased in size as will several of the other plots such as Figure 7.

---

## Author Comment (AC2) · 7 Feb 2018

Dear Referee #2

Many thanks for your comments and advice.

I have addressed all your questions below on behalf of my co-author as well, however I have not attached a modified manuscript yet. Please let me know if this is sufficient and the answer to a couple questions below.

1. The first is that no detailed description of the rotor or the diffuser is given so that future developers of DAWT models do not have the necessary information to test their theories.

We can include a figure each of the detailed geometry or a text description, namely that the rotor was of dimensions x and x and x, and the duct airfoil was an Eppler 423 with a chord of 0.6 m with inlet, throat and exit diameters of x, x, and x. Which would the referee feel to be more appropriate?

2. Secondly, the tests used a permanent magnet generator for which the manufacturer provides the efficiency curve for only one load. The experimental procedure of varying the load in search of the maximum power point is a sensible one, but it does not address the issue of the varying turbine efficiency. This is required to determine the extracted aerodynamic power, which is the usual target of the theories mentioned above.

Unfortunately, the efficiency of the generator was unknown. No load vs speed maps were provided from the manufacturer, despite requests, and we were simply told it was 95% efficient at rated output. Since the purpose of the test was to evaluate the potential of the design, and if the generator efficiencies were actually lower in reality, that would be supportive of the aero results, we simply varied the load in search of the best power point. Hence the Cp values represent the Cp for the turbine system, not the aero specifically. Do you have any suggestions as to best describe this in the text?

---

## Editor Comment (EC1) · J.ÂăN. Sørensen (Editor) · 8 Feb 2018

**J.ÂăN. Sørensen (Editor)**

jnso@dtu.dk

Both reviews are quite positive and suggest some minor revisions of the paper. I recommend that the authors upload a modified version of the paper for a final check of the reviewers.

---

## Author Response (AR1)

**Experimental Validation of a Ducted Wind Turbine Design Strategy**

Benjamin Kanya[1], Kenneth D. Visser[1]

[1]Department of Mechanical and Aeronautical Engineering, Clarkson University, Potsdam, NY, USA 13699

5     *Correspondence to*: Ken Visser (visser@clarkson.edu)

Note: Revised manuscript is attached following these comments to the reviewers and changes are noted in red

**Referee #1**

***What are the new findings and what are the inventions***

The primary finding is that a computational design strategy, involving the evaluation of a numerical solution of the ducted flow field as an input to the design code for the rotor itself, has been experimentally validated. This enables one to design a ducted turbine to a different size with a much higher degree of confidence in the real world performance. It also enables us to

15     achieve higher Cp values. These points were re-emphasized in the text and noted in the abstract and conclusions

***For the numerical method, what kind of grid type did the author employ?***

The following was added to the text:

*The grid had a boundary layer mesh with a growth rate of 1.1 and the first mesh point was set at $y^+ \approx 1$. The boundary*

20     *layer thickness was calculated as a function of $Re_c$ for each case and enough inflation layers were used to span the entire boundary layer. A layer of quadrilateral elements covered the actuator disc which was used to model the turbine in our 2D axisymmetric model. There was a refined unstructured triangular grid around duct which was surrounded by a large structured quadrilateral grid covering further the upstream and downstream of the actuator disc. The rest of domain was meshed with unstructured quadrilateral elements*

***What is the resolution around the turbine?***

The following was added to the text:

*The actuator disc is covered with 200 elements (i.e. for the 2.5m rotor each element in our axisymmetric model covered 6.25 mm)*

*Did the authors pay attentions to the Reynolds number effect and the grid resolution dependence to make clear the flow characteristics around the ducted wind turbine which shows flow separation and reattachment inside the duct, and vortex shedding from the duct.*

5    The following was added to the text:

*...the boundary layer thickness and y+ was calculated for each case to make sure the most accurate results that one can obtain from a RANS CFD solution can be obtained. The $k - \omega$ SST turbulence model was utilized, which among the two-equation turbulence model gives better prediction of flow separation.*

10   *The authors discussed the surface flow feature near the exit of the duct in the 4.2 Flow Field Issues. It strongly suggest that the flow around the ducted turbine both inside and outside of the duct are highly unsteady, unstable and turbulent flows. The reviewer cannot understand the accuracy of CFD presented in this paper.*

Although the predictions of power output from the CFD model agreed very well with the experimental results, as you suggested there was a difference between CFD predictions and experimental results. The following was added to the text:

15   *The CFD model showed no flow separation whereas flow separation was observed in the experimental tests. This could be due to simplifications of the CFD model like using a 2D axisymmetric model where the turbine was replaced with an actuator disc, or the limited accuracy of two-equation turbulence models. It could also have resulted from the differences of the manufactured model from the CFD model; from the final geometry of the duct, to the actual physical supporting structures used for the duct, and the influence of the individual rotor blades (as opposed to a uniform disc) which affect the*

20   *flow field. Lastly, the proximity of the ducted wind turbine to the floor could have aggravated the flow separation.*

*For the wind tunnel experiments, what is the uniformity of approaching wind in the wind tunnel of the University of Waterloo?*

The following was added to the text:

25   *The uniformity of the wind was not explicitly mapped. The flowfield is generated by a bank of 6 100 hp fans with independent variable speed control. Data was acquired with a single location sonic anemometer. At a given velocity setting the flowfield was sampled with a hand held anemometer and compared to the single point. The average variation of the incoming flow to the turbine varied by a +/- 1-2%.*

30   *What is the turbulence intensity and its uniformity?*

The following was added to the text:

*The turbulence of the flow filed was sampled over the range of the incoming velocities and was found to vary over the range of 5-10%*

*What is the blockage ratio?*

The following was added to the text:

*The blockage ratio, based on the projected frontal area of the duct and the rotor, for a stationary rotor, was 4.1%. If the rotating rotor is considered to act as a solid blockage, the ratio would be 11.7% No blockage corrections were made.*

*For the ducted wind turbine prototype employed here, the authors should describe the figure of the curved shape of the duct with the inlet diameter, throat diameter and exit diameter*

A figure of the detailed geometry of the duct was added to as Figure 8. Figures 7, 12 and 13 were deleted from the earlier copy as they were not value added.

*The referee recommends that the author should evaluate the modified Cp which adopts the maximum duct area, i.e., the projected area of the duct as the refer- ence area. For reference, please check a paper of Energies 2010, 3, 634–649; doi:10.3390/en3040634 "A Shrouded Wind Turbine Generating High Output Power with Wind-lens Technology"*

The following was added to the text:

15 *One possibility is to non-dimensionalize the data by the exit area of the duct. Scaling by the maximum projected area of the duct can be argued to be a more 'fair' evaluation of the data, when compared to a conventional open rotor turbine. In the case here, the exit diameter is 3.3 m and the rotor diameter is 2.5 m. Hence the Cp values would all be scaled by the ratio $2.5^2/3.3^2$ or a factor of 0.574. Cp values for the ducted configuration would then be in the range of 0.49 – 0.52, still better than an open rotor of this size.*

**Detailed points**

*1. p. 4, l 18-20; the author should explain "a" in more detail. And in also Figure 6, what are ωr(1+a'), Vo(1-a). Please describe the definitions of all the symbols in the present paper.*

A more detailed explanation of a and a', and the other variables was put in the manuscript. The following was added to the

25 text:

*… the factor by which the upstream flow velocity is slowed to by the time it reaches the rotor plane. For an ideal open rotor, a = 1/3 to maximize the power extracted. Other local variables at radius, r, to be noted include: Θ, the blade pitch angle; α, the angle of attack; ø, the angle the velocity vector, W, makes with the rotor plane; ω, the angular velocity; dQ, the elemental torque; dT, the elemental thrust; Vo, the upstream velocity; L, lift; D drag; and a', the angular velocity induction*

30 *factor.*

*2. Figure 5 b); the scales are written in so small letters*

The font size of the scales in Figure 5 were increased in size

**Referee #2**

*The first is that no detailed description of the rotor or the diffuser is given so that future developers of DAWT models do not have the necessary information to test their theories.*

5     Additional details of the geometry were added to the text including a drawing with dimensions, Figure 8

*Secondly, the tests used a permanent magnet generator for which the manufacturer provides the efficiency curve for only one load. The experimental procedure of varying the load in search of the maximum power point is a sensible one, but it does not address the issue of the varying turbine efficiency. This is required to determine the extracted aerodynamic*

10 *power, which is the usual target of the theories mentioned above.*

Unfortunately, the efficiency of the generator was unknown. No load vs speed maps were provided from the manufacturer, despite requests, and we were simply told it was 95% efficient at rated output. Since the purpose of the test was to evaluate the potential of the design, and if the generator efficiencies were actually lower in reality, that would be supportive of the aero results, we simply varied the load in search of the best power point. Hence the Cp values represent the Cp for the

15 turbine system, not the aero specifically. The aero efficiency would actually be higher, but no claim is made to that beyond system efficiency.

[revised manuscript text omitted]

5    **Figure 10: Aluminium Rotor Blade.**

[Figure]

**Figure 11: University of Waterloo Wind Turbine Test Facility.**

[Figure]

**Figure 12: Ducted Rotor Under Test Conditions.**

[Figure]

**Figure 13: Wind Tunnel Power Curve Performance.**

[Figure]

**Figure 14: Experimental Cp Values (based on rotor area).**

[Figure]

[Figure]

**Figure 15: Tufting on the Ducted Surface.**